# Overcoming challenges in the economic evaluation of interventions to optimise antibiotic use
Laurence S. J. Roope [1,2] ✉, Liz Morrell [1], James Buchanan[1,3], Alice Ledda[4,5], Amanda I. Adler [6], Mark Jit [7], A. Sarah Walker[2,5,8], Koen B. Pouwels [1,5,16], Julie V. Robotham[4,5,16], Sarah Wordsworth[1,2,5,16] & on behalf of the STEPUP team*

## Abstract

Bacteria are becoming increasingly resistant to antibiotics, reducing our ability to treat infections and threatening to undermine modern health care. Optimising antibiotic use is a key element in tackling the problem. Traditional economic evaluation methods do not capture many of the benefits from improved antibiotic use and the potential impact on resistance. Not capturing these benefits is a major obstacle to optimising antibiotic use, as it fails to incentivise the development and use of interventions to optimise the use of antibiotics and preserve their effectiveness (stewardship interventions). Estimates of the benefits of improving antibiotic use involve considerable uncertainty as they depend on the evolution of resistance and associated health outcomes and costs. Here we discuss how economic evaluation methods might be adapted, in the face of such uncertainties. We propose a threshold-based approach that estimates the minimum resistance-related costs that would need to be averted by an intervention to make it cost-effective. If it is probable that without the intervention costs will exceed the threshold then the intervention should be deemed cost-effective.

Driven by widespread antibiotic use, bacteria are becoming increasingly resistant to antibiotics. This is reducing our ability to treat infections and threatens to undermine modern health care and global public health. For instance, many invasive surgical procedures, from joint replacements to Caesarean sections, as well as immunosuppressive chemotherapy, would be substantially more dangerous without effective prophylactic antibiotics[1]. It is now recognised that tackling the threat requires addressing multiple challenges. These include unblocking the pipeline for new antibiotics and developing cost-effective diagnostic tests to determine whether antibiotics are needed[1,2]. Whilst there has been discussion of the need for better

incentives for innovation in the development of new antibiotics[1], better incentives are also needed to enable the introduction and use of interventions that aim to optimise the use of antibiotics and preserve their effectiveness, known as antibiotic stewardship interventions.

In this perspective, we discuss how current methods of economic evaluation might be adapted for interventions that affect antibiotic use, given the uncertainties over the evolution of resistance, and its associated health outcomes and costs. This includes a wide range of possible antibiotic stewardship interventions. The approach we propose can also be applied to interventions that affect the use of other antimicrobials.

[1]Health Economics Research Centre, Nuffield Department of Population Health, University of Oxford, Oxford, UK. [2]NIHR Oxford Biomedical Research Centre, John Radcliffe Hospital, University of Oxford, Oxford, UK. [3]Health Economics and Policy Research Unit, Wolfson Institute of Population Health, Queen Mary University of London, London, UK. [4]AMR Modelling and Evaluation, UK Health Security Agency, London, UK. [5]NIHR Health Protection Research Unit in Healthcare Associated Infections and Antimicrobial Resistance, University of Oxford, Oxford, UK. [6]Diabetes Trial Unit, Oxford Centre for Diabetes, Endocrinology and Metabolism, Oxford, UK. [7]Centre for Mathematical Modelling of Infectious Diseases, London School of Hygiene & Tropical Medicine, London, UK. [8]Nuffield Department of Medicine, University of Oxford, Oxford, UK. [16]These authors contributed equally: Koen B. Pouwels, Julie V. Robotham, Sarah Wordsworth. *A list of authors and their affiliations appears at the end of the paper. ✉e-mail: Laurence.roope@dph.ox.ac.uk

## The impact of all interventions that impact use of antibiotics must be considered

There are many potential types of interventions that could help to optimise antibiotic use. The introduction of diagnostic tests into clinical pathways, to help inform decisions as to whether antibiotics should be prescribed, are perhaps the most widely discussed, but there could also be important roles for others. These include vaccination (e.g., against influenza, which although a virus often leads to antibiotics being used); public information campaigns (e.g., about appropriate antibiotic use); and infection prevention and control strategies, including guidelines to reduce infection incidence and thus the need for antibiotics (e.g., hospital cleaning and mandatory mask wearing in certain contexts). Interventions potentially also include changes to prescribing guidelines, or to forms of governance (e.g., introducing a stewardship committee at national and/or healthcare facility levels). It is also important to recognise that in many low and middle income countries (LMICs) the lack of access to antibiotics is often a bigger problem than overuse. Thus, interventions to optimise antibiotic use may include some that aim to increase antibiotic use, where this is needed.

Unfortunately, as has long been recognised, the methods typically used in economic evaluations of such interventions do not adequately incorporate many of the benefits that optimising antibiotic use could bring, through the potentially beneficial impact on antibiotic resistance[3]. This is a major obstacle to incentivising the development and use of a wide range of interventions that aim to improve antibiotic use. Conversely, not adequately accounting for potential adverse impacts of antibiotic use on resistance is a barrier to comprehensive economic evaluation of interventions that aim to promote access to antibiotics in low-resource settings.

Many antibiotic optimisation interventions aim to tackle resistance by safely reducing antibiotic use, and for illustrative purposes we will focus mainly on these. However, it is important to recognise that resistance mechanisms are varied and complex, and that optimising antibiotic use does not necessarily entail reducing use in all circumstances. Optimising antibiotic use can be just as much about choice of antibiotic, as well as appropriate dose and duration. It may even involve increasing antibiotic use in some circumstances. For example, there is evidence that, in some situations, targeted increases in use of a specific antibiotic, informed by diagnostics, could lead to a decrease in resistance, by reducing the opportunities for transmission of resistant strains[4]. Increased, rather than decreased, use can also sometimes be optimal for other types of antimicrobials. For example, combination therapy is typically used during antiretroviral therapy in people infected with human immunodeficiency virus (HIV), partly to help to prevent within-host evolution of drug resistance[5]. Poor adherence to therapy may lead to increased resistance[6]. Whilst we focus our discussion on interventions to reduce antibiotic use, the methods we propose can equally be applied to any interventions that aim to optimise the use of either antibiotics or other antimicrobials.

## Measuring the benefits of antibiotic optimisation

In recent years, there have been various attempts to estimate the impact of antibiotic use on resistance and incorporate this into economic evaluation. For example, economic evaluations of stewardship interventions have been conducted that simulate the evolution of resistance and its impact on health and economic outcomes, with and without an intervention[7]. However, all attempts to estimate the broad benefits from antibiotic optimisation are subject to considerable uncertainty over the evolution of resistance, and its associated health outcomes and costs.

## Adapting economic evaluation methods to evaluate interventions to improve antibiotic use

While better access to existing antibiotics remains a central challenge in many low-income countries, worldwide there is clearly a need to reduce unnecessary antibiotic use. A key barrier to the development and use of new stewardship interventions, particularly diagnostic tests, has been the difficulty in demonstrating their value, which requires comprehensive economic evaluations. If they are found to be cost-effective, this is likely to facilitate favourable 'reimbursement' decisions, where the payer (e.g., health insurer or government) agrees to reimburse the technology provider at a given price.

Initially, improved economic evaluation of interventions to optimise antibiotic use is likely to mainly affect the availability and use of these interventions in high-income countries. Yet, in terms of resistance globally, interventions to optimise antibiotic use in LMICs could ultimately have the greatest impact. There are multiple challenges in many LMIC contexts with implementing stewardship interventions, including lack of capacity for microbiological testing and availability of antibiotics without prescriptions outside the formal healthcare system[8]. If these challenges can be overcome, the methods of economic evaluation we propose here could also be applied in LMICs. However, in the meantime, a positive economic evaluation, and economic viability in high-income settings, can potentially be a first step towards eventual availability also in LMIC settings. It has been suggested that the use of effective diagnostics in the latter settings should be supported via a globally administered diagnostic market stimulus system, in which direct subsidies are provided to diagnostic test manufacturers on a per use or purchase basis[2].

While it is relatively easy to estimate the costs of many antibiotic stewardship interventions, quantifying their potential benefits is extremely challenging. This is mainly because it is complicated to estimate how changes in antibiotic use will affect long-term incidence and the resulting costs of antibiotic resistance. Figure 1 illustrates how antibiotic consumption can increase the overall costs of treating infections as a result of the development and spread of antibiotic resistance. Table 1 summarises key elements that pose a challenge in quantifying these links.

Standard methods for economic evaluation of health technologies usually focus on evaluating the benefits of interventions to individual patients, for example a reduction in the duration of symptoms, relative to their incremental costs. In the case of antibiotics, particularly those widely used in the community, such as amoxicillin for respiratory tract infections, treatment costs are generally far lower than the cost of diagnosis. This results in antibiotics being used extensively as a precaution in situations when they might not be needed. There is a need to evaluate interventions in a way that also captures the broader benefits they might bring by reducing the overuse of antibiotics that drives the emergence, establishment and transmission of antibiotic-resistant pathogens. Without capturing these broader benefits it is likely that the suboptimal use of antibiotics will continue.

Without effective antibiotics to prevent or treat infections that are a consequence of antibiotic resistant pathogens, there are likely to be more frequent outbreaks of difficult-to-treat infections. Many medical treatments and procedures need antibiotics to be administered as prophylaxis, for example joint replacements and Caesarean sections. These procedures will become riskier, or even unsafe. Furthermore, substantial increases in the occurrence of resistant bacterial infections might lead to more frequent and longer hospital stays, which has a wider impact on a patient's health and increases hospital costs. This can have an even wider impact if a consequence is insufficient beds or health worker availability for patients requiring other types of care. These are all potential benefits of having effective antibiotics available that, with rare exceptions[7,9], health economic evaluations generally do not consider.

Most existing antibiotics are relatively cheap and can bring tangible short-term benefits, at least in those with bacterial infections. Conversely, the (long-term) costs of increased resistance to antibiotics, driven by antibiotic use, may be large but are difficult to quantify. This difficulty is largely due to challenges in predicting the development and spread of antibiotic resistance genes under alternative antibiotic use scenarios. An additional reason is that there is a lack of high-quality data on economic outcomes, some of which cannot be directly observed, such as the cost of being unable to perform invasive surgery[10]. Consequently, the latter costs are typically absent from economic evaluations of interventions that could improve antibiotic use. The result is that the economic evaluations that drive health care reimbursement decisions might incorrectly conclude that possible interventions to improve antibiotic use are not cost-effective, particularly for more expensive interventions. There are similar challenges in the economic

**Fig. 1 | Links between incremental antibiotic use and additional costs of treating resistant infections.** In evaluating the cost-effectiveness of interventions to improve antibiotic use, it is important to find a way to account for the various pathways via which incremental antibiotic use might affect costs due to increased resistance. This figure features a number of pathways via which incremental antibiotic use can increase the cost of treating infections. Economic evaluations of interventions to optimise antibiotic use should account for all these pathways; not only the more easily parameterised pathways in which antibiotic treatment reduces costs by reducing transmission of susceptible bacteria. The arrow to de novo resistance through mutagenesis is dashed, because while this route is likely important for tuberculosis, gonorrhoea and specific bug-drug combinations, other routes might be more important for the majority of infections with resistant bacteria.

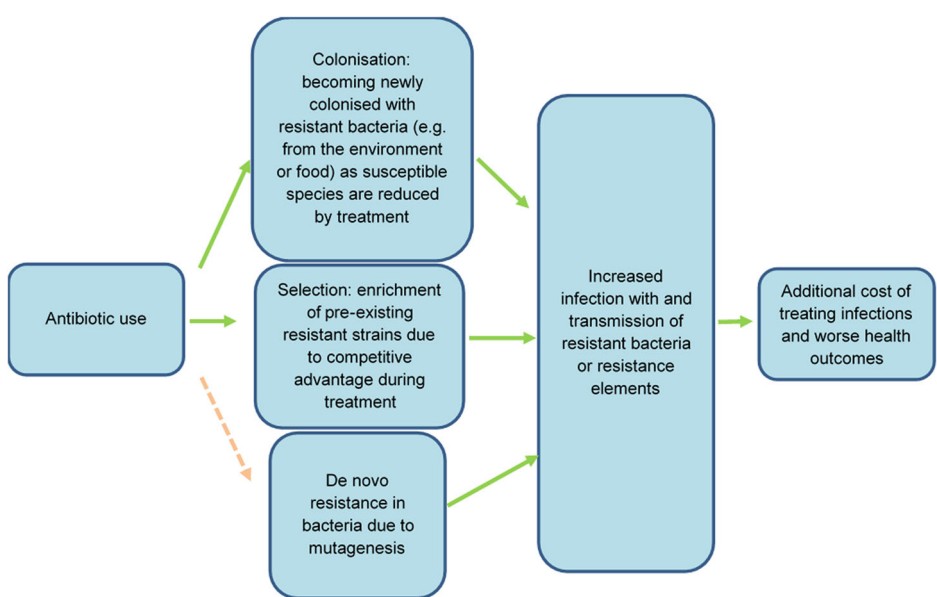

**Table 1 | Summary of key elements that pose a challenge in economic evaluation of stewardship interventions**

| Elements | Comments |
|---|---|
| Impact of antibiotic use on spread of resistance | Resistant strains are typically enriched during antibiotic treatment, due to their competitive advantage over susceptible strains. However, the extent of this enrichment, for a given increase in antibiotic use, varies by bug-drug combination and is difficult to predict. |
| Extent to which resistance can be reversed by reducing antibiotic use | This probably depends on a variety of factors, including the fitness cost of the resistance mechanism, epidemic potential of bacteria/strain, cross-resistance with alternative antibiotics, and environmental considerations. |
| Effects of antibiotic use on susceptibility to colonisation with resistant bacteria | The impact of antibiotic use on subsequent risk of infection with resistant bacteria, and the associated health-economic outcomes, is poorly understood. |
| Lack of high-quality data on economic outcomes | This is especially problematic in the context of events without precedent, such as costs from being unable to perform invasive surgery if effective prophylactic antibiotics become unavailable, or if infections become substantially harder to treat than they have been previously. |
| Impact of antibiotic use on emergence of resistance in bacteria | While de novo resistance in bacteria, due to mutagenesis, may not be relevant for most infections with resistant bacteria, it is probably important for tuberculosis, gonorrhoea and specific bug-drug combinations. Predicting the emergence of resistance is notoriously difficult. |

evaluation of novel antibiotics, where there have been recent calls for health technology assessment (HTA) agencies to broaden their methodological tool kit and incorporate long-term goals, such as containing resistance, as part of their evaluation criteria for new antibiotics[11]. While this is a different context, in that optimising antibiotic use is not the goal, a common theme is the need to value and incentivise future access to effective antibiotics by considering how antibiotic use affects resistance.

## Incorporating long-term costs of antibiotic resistance is complicated

There are similarities between the challenge of addressing antibiotic resistance and climate change. The feasibility of applying methods from the economics of climate change to evaluate interventions to improve antibiotic use have been discussed[1]. Similarly to cost-benefit analyses of carbon emissions, in principle, cost-benefit analyses of use of a given antibiotic could be conducted by monetising all the current and future costs and benefits of this use, and discounting them over time, to obtain the net present value (NPV) of antibiotic consumption. This approach would be especially attractive if current antibiotic use has a long-lasting impact on future costs, e.g., via increased rates of resistance years or decades ahead. Note that, in contrast to carbon, there are many types of antibiotics, and the NPV will be expected to vary by antibiotic type and targeted organism. For example, despite high levels of nitrofurantoin use to effectively treat uncomplicated urinary tract infections, nitrofurantoin resistance among

*Escherichia coli* remains low[12]. As noted above, in some situations, targeted increases in use of a specific antibiotic could lead to a decrease in resistance[4].

Evidence on how quickly and how much antibiotic resistance can be reversed is mixed and depends on several factors. These include the fitness cost of the resistance mechanism, i.e., the extent to which susceptible bacteria outcompete resistant bacteria in an environment without antibiotics, the concentration range at which antibiotics select for resistance[12], the epidemic potential of the bacteria/strain, transmission routes, and the presence of environmental sources providing selective pressure. Importantly, just replacing one antibiotic with another, without also reducing use, might not reduce resistance for various reasons. These reasons include cross-resistance between antibiotics; several resistance genes being transmitted together or selected for due to exposure to an antibiotic from a different class, due to their co-location in the genetic material; and increases in susceptibility to becoming colonised with resistant bacteria after antibiotic exposure[13,14].

Previous work emphasises the complexity of predicting even short-term changes in resistance (due to antibiotic use), with expected, short-term, relationships between antibiotic use and resistance demonstrated for some bug-drug combinations, but not others[15,16]. As long-term predictions are hard even for specific bug-drug combinations[17], cost-benefit analyses incorporating broad long-term costs and outcomes are extremely difficult. However, it is widely recognised that we need to find better pragmatic decision aids that do not ignore the wider and potentially large

costs of resistance and the benefits of having effective antibiotics in the future[11,18,19].

## Learning from methods used in the economic evaluation of newly available antibiotics

The economic evaluation of newly developed antibiotics requires similar evaluations, and there have been recent developments introduced in an effort to incentivise the development of new antibiotics. The UK's National Institute for Health and Care Excellence (NICE) has introduced a pilot subscription-based approach to reward drug developers for developing new antibiotics. In the pilot approach, the National Health Service (NHS) purchases antimicrobials on a subscription basis in which they pay a set price for them to be available, rather than the payment amount being based on the volumes of antibiotics sold. It is hoped this will incentivise innovation[1] because the manufacturer will be paid regularly based on the estimated benefit the drug offers, regardless of the volume supplied. In another recent pilot, the Public Health Agency of Sweden (PHAS) introduced a scheme where rewards for supplying several novel antibiotics were partially delinked from volumes sold. In this case, the drugs were paid for by the state on a per use basis, but the state provided a minimum guaranteed payment, in the event that the volumes sold were below a certain level, as well as offering a bonus should there be an unexpectedly large volume of sales[20]. Beyond the UK and Sweden, progress on delinking rewards for antibiotic innovation from volumes remains very limited[20].

The appraisal of the drugs in the UK pilot (cefiderocol[18] and ceftazidime–avibactam[19]) follows a new methodological approach recommended by the Policy Research Unit in Economic Methods of Evaluation in Health and Social Care Interventions[21]. This considers broad societal benefits of having effective antibiotics, and is characterised by: output of a total long-term value estimate for the drug to inform the subscription price negotiation, rather than a yes/no decision at a proffered price; value presented as an overall population-level net health benefit, in Quality Adjusted Life Years (QALY), rather than the usual incremental patient-level cost-per-QALY; high reliance on expert opinion to project future patterns of infection and resistance, notably from a committee selected specifically for their expertise; and explicit incorporation of additional benefits not otherwise captured in the model (e.g., having access to a more diverse set of antibiotics), via multipliers on the QALY gain.

Whilst the methodology directly supports the subscription approach and is a step forward, it does not overcome the challenges of modelling the evolution of resistance with antimicrobial usage[15,16], or the difficulties in adequately incorporating the broad benefits of the new drugs. These factors are dealt with using multipliers based on expert input, which are inevitably subjective. Furthermore, the level of uncertainty is increased by the use of in vitro data to model clinical outcomes. Despite these limitations, the approach could form the basis of research prioritisation using value-of-information analyses to decrease uncertainty and reliance on expert opinion for influential parameters[22].

The approach we propose for the economic evaluation of interventions to improve antibiotic use has some points of commonality with NICE's methodology for evaluating new antibiotics, such as focusing on broad population level costs and benefits. However, for new antibiotics, the key issue is the benefits (whether in QALYs or monetary terms) that a new drug will bring; for interventions to improve antibiotic use, a key concern is the cost (whether in QALYs or monetary terms) of not optimising antibiotic use.

## Economic evaluation of antibiotic interventions based on probability of costs exceeding a threshold

There is an extensive literature, beyond health care, on how to conduct economic evaluations when costs or benefits are difficult to quantify, or include resources deemed to have value simply by being available, even if they are not necessarily used (non-use or option value). For example, revealed and stated preference methods are widely used to place a monetary value on environmental resources[23]. Consider a situation where an antibiotic

stewardship intervention, with potential to reduce some percentage of unnecessary use of some relatively cheap antibiotic, is being evaluated. Suppose that, partly due to its high up-front costs and the relatively much lower cost of antibiotics, an intervention would only be deemed cost-effective if it brings sufficient future benefits from reduced costs of resistance. A pragmatic approach would be to start by estimating the minimum costs (from resistance) that would need to be averted by this stewardship intervention's reduction in antibiotic doses, to make the intervention favourable, i.e., the intervention scenario resulting in a positive net monetary benefit, net health benefit, or NPV. We view this as a cost threshold, above which an intervention would be cost-effective. A similar approach has been advocated previously in the context of evaluating whether use of an antibiotic to treat an infection is cost-effective; however, such approaches have not been used in antibiotic stewardship interventions and are very rarely used in any area of HTA[24]. Importantly, while similar methods might be used across varying country and health system contexts, the results of any economic evaluation of interventions to optimise antibiotic use are likely to be highly context specific. Thus, to inform national decision making, models should be populated, as far as possible, with the most relevant, country-specific data available.

In the absence of good evidence on how likely exceeding this extra cost threshold is, we could estimate it. These estimates could be informed by eliciting the views of clinical and infectious disease experts[11,19] on the extent to which reduced antibiotic use from a given intervention would be likely to lower resistance levels. To avoid potential biases, it would be important to adhere to established guidelines for selecting experts and eliciting their preferences[25,26]. If the probability, without the intervention, of costs exceeding the threshold is high (say, greater than some predetermined probability $p$), the intervention should be deemed 'cost-effective'. Setting a sensible value for this probability $p$ will in itself be an important decision. Considering both the precautionary principle and the possibility of extremely high costs in future (even if occurring with very small probability), a good choice would probably be well below 0.5. Lower values for $p$ might also be argued for on the basis that access to effective antimicrobials can bring a broad range of economic benefits, well beyond the most obvious health-care related costs; these are difficult to quantify but likely to be substantial[27]. A more involved cost-benefit analysis approach could involve estimating the probabilities of different cost levels at different time horizons and using these to estimate the NPV of the intervention, or the probability of the NPV being positive, where a positive NPV indicates adoption.

## Outlook

Without adequate consideration of the future benefits from reduced resistance, there is a risk that policymakers could use economic evaluation of antibiotic optimisation interventions as justification for disinvestment rather than investment. Consider a situation where data on costs and benefits are limited, especially with regard to costs of future resistance avoided by an intervention. A traditional economic evaluation might omit these future resistance-related benefits completely, with the probable consequence of finding optimisation interventions not cost-effective. Our approach discourages ignoring these future benefits by demanding a consideration, in such circumstances, of the minimum future resistance-related benefit (i.e., the threshold) that would be needed from the intervention to make the conclusion of the analysis switch to being cost-effective, rather than not cost-effective.

In this way, economic evaluations of antibiotic optimisation interventions could be obliged to include a reasonable attempt to estimate the probabilities of exceeding this extra cost threshold, or the probability of a positive NPV. Initially, these estimates will probably need to rely largely on information elicited from experts. Over time, as evidence from randomised trials, observational analyses, transmission models and our understanding of resistance emergence and growth improves, our estimates of probabilities of exceeding cost thresholds will improve and we can employ more definitive evidence. For example, there should soon be evidence on the effects of antibiotic use on susceptibility to colonisation with resistant bacteria[14], and

on the subsequent risk of infection and associated health-economic outcomes. Incorporating these outcomes in randomised trials evaluating the impact of interventions to improve antibiotic use could greatly improve our understanding of the link between changes in prescribing, antibiotic resistant infections, and subsequent health-economic outcomes.

However, we need to do more with the existing evidence. The available evidence on the current economic burden of antibiotic resistant infections could already be incorporated into models predicting the health economic impact of interventions to prevent such infections[10]. Importantly, various randomised trials have received funding to address some key gaps in the current health-economic evidence in this area, such as pREVention and management tools for rEducing antibiotic Resistance in high prevalence Settings (REVERSE)[28], and DURATION OF ANTIBIOTIC TREATMENT IN URINARY TRACT INFECTION (DuRATIon-UTI)[29]. There is much to be learned that can help us refine economic evaluations of antibiotic optimisation interventions, building on the extra cost threshold type of evaluation that we propose. However, we believe enough is already known to start using them now and the urgent need to improve the way that we use antibiotics highlights the need to do so without delay.

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

## Acknowledgements

The study was funded by the Economic and Social Research Council (ESRC) through the Antimicrobial Resistance Cross Council Initiative supported by the seven research councils in partnership with other funders (grant reference: ES/P008232/1). L.S.J.R., A.I.A., A.S.W., and S.W. are supported by the NIHR Oxford Biomedical Research Centre, Oxford. A.S.W. is a NIHR Senior Investigator. The views expressed are those of the author(s) and not

necessarily those of the NHS, the NIHR, or the Department of Health and Social Care.

## Author contributions
L.S.J.R. led the drafting of the manuscript, with contributions from L.M., J.B., A.L., A.I.A., M.J., A.S.W., K.B.P., J.V.R., and S.W. L.S.J.R., L.M., J.B., A.L., A.I.A., M.J., A.S.W., K.B.P., J.V.R., and S.W. read and approved the final manuscript.

## Competing interests
The authors declare no competing interests.

## Additional information

---

## on behalf of the STEPUP team

**Philip E. Anyanwu**[9], **Aleksandra J. Borek**[10], **Nicole Bright**[8], **James Buchanan**[1,3], **Christopher C. Butler**[10], **Anne Campbell**[11], **Céire Costelloe**[11], **Benedict Hayhoe**[12], **Alison Holmes**[13], **Susan Hopkins**[14], **Azeem Majeed**[12], **Monsey McLeod**[11], **Michael Moore**[15], **Liz Morrell** ⓘ[1], **Koen B. Pouwels** ⓘ[1,5,16], **Julie V. Robotham**[4,5,16], **Laurence S. J. Roope** ⓘ[1,2] ✉, **Sarah Tonkin-Crine**[10], **A. Sarah Walker**[2,5,8], **Sarah Wordsworth**[1,2,5,16], **Carla Wright**[8], **Sara Yadav**[13] & **Anna Zalevski**[8]

[9]Warwick Medical School, University of Warwick, Coventry, UK. [10]Nuffield Department of Primary Care Health Sciences, University of Oxford, Oxford, UK. [11]National Institute for Health Research (NIHR) Health Protection Research Unit in Healthcare Associated Infections and Antimicrobial Resistance, Imperial College London, London, UK. [12]Department of Primary Care and Public Health, School of Public Health, Imperial College London, London, United Kingdom. [13]Department of Infectious Disease, Impe, rial College London London, UK. [14]Clinical and Public Health Group, UK Health Security Agency, London, UK. [15]Primary Care, Population Sciences and Medical Education, Faculty of Medicine, University of Southampton, Southampton, UK.

