## [Peer Review File · Communications Medicine]

Reviewers' comments:

Reviewer #1 (Remarks to the Author):

Thank you for submitting this manuscript. I have no negative comment on it. It is well-composed, with impactful insight.

Reviewer #2 (Remarks to the Author):

The authors explore the challenges of incorporating broad and long-term impact of reducing AMR in the framework of cost-effectiveness analysis of antimicrobial stewardship interventions. They propose a view of the AMR pathways and suggest solutions introducing the NPV approach.

Please add line numbers to facilitate referring to specific parts during review.

The title is not comprehensive of the scope of this Perspective as it only mentions the reduction of unnecessary use, whereas stewardship has more dimensions (e.g. duration, appropriateness etc).

Antibiotic stewardship encompasses all interventions aiming at providing the right antibiotic with the right dosage to the right persons for the right duration. Hence, the aim is not necessarily to reduce use, but to ensure correct/appropriate use.

The authors should also consider the importance of prescription guidelines (and more generally for infection care), as well as governance e.g. having a stewardship committee (and how this could be integrated with IPC committees) at national and healthcare facility levels

I'm not sure it's fair to state that since 2002 (the reference chosen by the authors which I find outdated) there has not been interesting and successful attempts to estimate the benefits stemming from the cited interventions, one example being the OECD Stemming the superbug tide. The very recent update also includes much more evidence and modelling on the indirect costs of AMR.

The exposition of the elements that pose a challenge in cost-effectiveness analysis of stewardship would benefit from a summary table outlining them (from those with short to those with long-term impact of AMR), e.g. cost of not performing routine care, but also fitness cost and resistance reversal.

The NPV mentioned by the authors would probably be specific to bug-drug combinations and to an epidemiological context (not only to the intervention tested); the Perspective would benefit from the authors touching upon this.

Authors' Response to Reviewer Comments

Reviewers' comments:

Reviewer #1 (Remarks to the Author):

Thank you for submitting this manuscript. I have no negative comment on it. It is well-composed, with impactful insight.

Reply: We appreciate the reviewer's positive comments.

Reviewer #2 (Remarks to the Author):

The authors explore the challenges of incorporating broad and long-term impact of reducing AMR in the framework of cost-effectiveness analysis of antimicrobial stewardship interventions. They propose a view of the AMR pathways and suggest solutions introducing the NPV approach. Please add line numbers to facilitate referring to specific parts during review.

Reply: We have added line numbers as requested.

The title is not comprehensive of the scope of this Perspective as it only mentions the reduction of unnecessary use, whereas stewardship has more dimensions (e.g. duration, appropriateness etc). Antibiotic stewardship encompasses all interventions aiming at providing the right antibiotic with the right dosage to the right persons for the right duration. Hence, the aim is not necessarily to reduce use, but to ensure correct/appropriate use.

Reply: We agree with the reviewer that stewardship has more dimensions than simply reducing unnecessary use. As suggested, we have amended the title to refer to "optimising antibiotic use" rather than only reducing "unnecessary antibiotic use." The amended title is: "Overcoming challenges in the economic evaluation of interventions to optimise antibiotic use"

The authors should also consider the importance of prescription guidelines (and more generally for infection care), as well as governance e.g. having a stewardship committee (and how this could be integrated with IPC committees) at national and healthcare facility levels

Reply: We have expanded the discussion of antibiotic related interventions in the introductory paragraph to include these examples.

I'm not sure it's fair to state that since 2002 (the reference chosen by the authors which I find outdated) there has not been interesting and successful attempts to estimate the benefits stemming from the cited interventions, one example being the OECD Stemming the superbug tide. The very recent update also includes much more evidence and modelling on the indirect costs of AMR.

Reply: We have added several lines discussing the approach of the OECD report flagged by the reviewer – thank-you. We have also added that, despite such developments, all attempts to estimate the broad benefits from reduced antibiotic use are inevitably subject to considerable uncertainty over the evolution of resistance, and its associated health outcomes and costs. It is against this backdrop of uncertainty that we go on to propose the perspective’s threshold-based approach. (See lines 45-52 in untracked version)

The exposition of the elements that pose a challenge in cost-effectiveness analysis of stewardship would benefit from a summary table outlining them (from those with short to those with long-term impact of AMR), e.g. cost of not performing routine care, but also fitness cost and resistance reversal.

Reply: As suggested, we have provided a summary table in the revised manuscript that outlines these challenges (Table 1).

The NPV mentioned by the authors would probably be specific to bug-drug combinations and to an epidemiological context (not only to the intervention tested); the Perspective would benefit from the authors touching upon this.

Reply: We agree that the NPV of an intervention would be specific to the particular antibiotic(s), and the organism(s), that the stewardship intervention is targeting. In the revised manuscript, we have therefore added text (lines 134-140 and line 221 in untracked version) clarifying that the NPV is conditional on the type of antibiotic and targeted organism.

Reviewers' comments:

Reviewer #2 (Remarks to the Author):

Thank you for the revision and tracked changed word document on which I worked.

In the first page, the authors acknowledge that AMR is not only a problem related to overuse of antibiotics, but then already starting at the end of page 1 they quickly revert to focusing on overuse as main component to consider when undergoing economic evaluations (also Table 1 and the elements that pose a challenge). We still do not understand the reason for this focus, please explain or change to optimising use. To note, that it's the latter that will have an impact on AMR, not necessarily only reducing use (this is also acknowledged by the authors' use of reference 10).

Reviewer #3 (Remarks to the Author):

-Generally the manuscript is an important addition to the literature and will help stimulate debate around how to more routinely incorporate cost-effective analyses (CEA) in the evaluation of AMR interventions. I have a few specific points the author may wish to consider:

-It appears that authors focus exclusively on antibiotics, but much of the discussion is relevant to antimicrobials generally, therefore the authors may wish to explicitly state they are only concerned with antibiotics from the beginning, or acknowledge these principles are applicable to antimicrobials
-LMICs- do the authors wish to discuss whether CEAs methods and data collection need to be further developed that are applicable to LMICs? Rather than working on the assumption that CEA conducted in high-income countries will automatically promote availability of stewardship interventions in LMICs. If we are taking a more global perspective, increased attention to establishing what is cost-effective in LMICs for AMR may have a greater impact on AMR.

-The authors also need to acknowledge that estimating NPV in one health system or country context may not be generalisable to other countries, and that there may be benefit in development of the frameworks/methods to deploy this approach which can be populated with more country specific data to support national decision making

-The authors need to discuss some of the potential dangers of this approach, in general I am quite supportive of this, and I do believe there needs to be more routine CEA of AMR interventions, however there is a risk that mandating a specific "threshold" for NPV may steer investment only towards interventions where robust data exists on costs and benefits. In the context of scarcity, there is the risk that policy-makers will use CEA as justification for disinvestment rather than investment.

-In terms of the Figure, the pathways of colonisation, selection, and de novo resistance are sensible. The only point I wish to add is whether authors may wish to discuss whether their proposed approach to CEA is actually fairly conservative, as they do not include many of the broader economic benefits of effective antimicrobials outlined by Rothery et al 2018 ie "STEDI" factors (spectrum, transmission, enablement, diversity and insurance). While originally developed to establish value of novel antimicrobials, they are also applicable to other AMR interventions. Although the drawback of these factors are that they are often difficult to estimate and include within economic models (without significant uncertainty)

Authors' Response to Reviewer Comments

We have responded below to all reviewer comments. Please note that all page and line numbers mentioned here refer to the tracked version of the manuscript.

Reviewers' comments:

Reviewer #2 (Remarks to the Author):

Thank you for the revision and tracked changed word document on which I worked.

In the first page, the authors acknowledge that AMR is not only a problem related to overuse of antibiotics, but then already starting at the end of page 1 they quickly revert to focusing on overuse as main component to consider when undergoing economic evaluations (also Table 1 and the elements that pose a challenge). We still do not understand the reason for this focus, please explain or change to optimising use. To note, that it's the latter that will have an impact on AMR, not necessarily only reducing use (this is also acknowledged by the authors' use of reference 10).

Reply: As the reviewer notes, in the previous version of the manuscript, while acknowledging the importance of optimisation of antibiotics, we had focused primarily on overuse. We have changed the language throughout most of the revised manuscript to focus on "optimising" rather than "reducing". We have also substantially expanded the second paragraph in the manuscript with new text explaining that optimising antibiotic use (and indeed antimicrobial use more generally) is not always about reducing use:

"Many antibiotic optimisation interventions aim to tackle resistance by safely reducing antibiotic use, and for illustrative purposes we will focus mainly on these. However, it is important to recognise that resistance mechanisms are varied and complex, and that optimising antibiotic use does not necessarily entail reducing use in all circumstances. Optimising antibiotic use can be just as much about choice of antibiotic, as well as appropriate dose and duration. It may even involve increasing antibiotic use in some circumstances. For example, there is evidence that, in some situations, targeted increases in use of a specific antibiotic, informed by diagnostics, could lead to a decrease in resistance, by reducing resistant strains' opportunities for transmission [4]. Increased, rather than decreased, use can also sometimes be optimal in other types of antimicrobials. A notable example is in antiretroviral therapy for human immunodeficiency viruses (HIV). In HIV treatment, combination therapy is typical, partly to help to prevent within-host evolution of drug resistance [5], and poor adherence to therapy may lead to increased resistance [6]. Whilst we anchor the following discussion with reference to interventions around reducing antibiotic use, the methods we propose can equally be applied to any interventions that aim to optimise the use of either antibiotics or other antimicrobials."

References

[4] McAdams D, Wollein Waldetoft K, Tedijanto C, Lipsitch M, Brown SP. Resistance diagnostics as a public health tool to combat antibiotic resistance: A model-based evaluation. PLoS biology. 2019 May 16;17(5):e3000250.

[5] Broder S. The development of antiretroviral therapy and its impact on the HIV-1/AIDS pandemic. *Antiviral research*. 2010 Jan 1;85(1):1-8.

[6] Bangsberg DR, Kroetz DL, Deeks SG. Adherence-resistance relationships to combination HIV antiretroviral therapy. *Current HIV/AIDS Reports*. 2007 May;4(2):65-72.

Reviewer #3 (Remarks to the Author):

-Generally the manuscript is an important addition to the literature and will help stimulate debate around how to more routinely incorporate cost-effective analyses (CEA) in the evaluation of AMR interventions.

Reply: Thank-you for these positive comments.

I have a few specific points the author may wish to consider:

-It appears that authors focus exclusively on antibiotics, but much of the discussion is relevant to antimicrobials generally, therefore the authors may wish to explicitly state they are only concerned with antibiotics from the beginning, or acknowledge these principles are applicable to antimicrobials

Reply: Thank-you for raising this. In the revised manuscript, while we still anchor the discussions mainly with reference to interventions around optimising antibiotic use, we have attempted to broaden the discussion by explaining that our proposed methods are also applicable to any interventions that aim to optimise use of antimicrobial more generally. For example, the second paragraph in the manuscript now contains this text:

“Increased, rather than decreased, use can also sometimes be optimal in other types of antimicrobials. A notable example is in antiretroviral therapy for human immunodeficiency viruses (HIV). In HIV treatment, combination therapy is typical, partly to help to prevent within-host evolution of drug resistance [5], and poor adherence to therapy may lead to increased resistance [6]. Whilst we anchor the following discussion with reference to interventions around reducing antibiotic use, the methods we propose can equally be applied to any interventions that aim to optimise the use of either antibiotics or other antimicrobials.”

-LMICs- do the authors wish to discuss whether CEAs methods and data collection need to be further developed that are applicable to LMICs? Rather than working on the assumption that CEA conducted in high-income countries will automatically promote availability of stewardship interventions in LMICs. If we are taking a more global perspective, increased attention to establishing what is cost-effective in LMICs for AMR may have a greater impact on AMR.

Reply: We have expanded the second paragraph on page 3 to point out that “in terms of resistance globally, interventions to optimise antibiotic use in LMICs could ultimately have the greatest impact” (lines 90-91). We have explained that, despite the challenges of implementing antibiotic optimisation interventions in LMICs (including limitations in collecting high-quality data), the same methods of economic evaluation are actually applicable in LMICs (lines 95-96). We did not mean to suggest that conducting CEA in high-income countries would automatically promote availability of stewardship interventions in LMICs, only that it could potentially be a first step. We have amended the wording to make this clearer in the revised manuscript.

-The authors also need to acknowledge that estimating NPV in one health system or country context may not be generalisable to other countries, and that there may be benefit in development of the frameworks/methods to deploy this approach which can be populated with more country specific data to support national decision making

Reply: Thank-you for this suggestion. In the revised manuscript, we have addressed this by adding the following text (page 9, lines 263-268):

“Importantly, while similar methods might be used across varying country and health system contexts, the results of any economic evaluation of interventions to optimise antibiotic use – e.g. in terms of net monetary benefit, net health benefit, or NPV - are likely to be highly context specific. Thus, to inform national decision making, models should be populated, as far as possible, with the most relevant, country-specific data available.”

-The authors need to discuss some of the potential dangers of this approach, in general I am quite supportive of this, and I do believe there needs to be more routine CEA of AMR interventions, however there is a risk that mandating a specific “threshold” for NPV may steer investment only towards interventions where robust data exists on costs and benefits. In the context of scarcity, there is the risk that policy-makers will use CEA as justification for disinvestment rather than investment.

Reply: We are grateful to the reviewer for raising an important issue. In our view, one of the major advantages of the threshold-based approach we advocate is that it should actually reduce the risk that policy-makers would use CEA as justification for disinvestment rather than investment. Consider a situation where data on costs and benefits are limited, especially with regard to costs of future resistance avoided by an intervention. A traditional CEA might omit these future resistance-related benefits completely, with the likely consequence of finding optimisation interventions not cost-effective. Our approach discourages ignoring these future benefits by demanding a consideration, in such circumstances, of the minimum future resistance-related benefit that would be needed from the intervention to make the conclusion of the analysis ‘switch’ to being cost-effective, rather than not cost-effective. We have added a discussion along these lines as an extra paragraph at the start of the Outlook section (p. 10, lines 290-300).

-In terms of the Figure, the pathways of colonisation, selection, and de novo resistance are sensible. The only point I wish to add is whether authors may wish to discuss whether their proposed approach to CEA is actually fairly conservative, as they do not include many of the broader economic benefits of effective antimicrobials outlined by Rothery et al 2018 ie “STEDI” factors (spectrum, transmission, enablement, diversity and insurance). While originally developed to establish value of novel antimicrobials, they are also applicable to other AMR interventions. Although the drawback of these factors are that they are often difficult to estimate and include within economic models (without significant uncertainty)

Reply: We agree with the reviewer that there are broad benefits from effective antimicrobials. Our approach is sufficiently flexible to allow for varying levels of conservativeness to be embedded in

economic evaluations. This flexibility is built into the choice of probability p , i.e. the probability that the resistance-related benefits / costs avoided by the intervention are deemed to be above the threshold that would make the intervention 'switch' to being cost-effective. We have added the following text and reference to in our discussion about the choice of p (p. 9, lines 280-283).

"Higher values for p might also be argued for on the basis that access to effective antimicrobials can bring a broad range of economic benefits, well beyond the most obvious health-care related costs; these are difficult to quantify but likely to be substantial [27]."

Reference [27]: Brassel S, Firth I, Chowdhury S, Hampson G, Steuten L. Capturing the Broader Value of Antibiotics. Office of Health Economics; 2023 Nov 23.

REVIEWERS' COMMENTS:

Reviewer #2 (Remarks to the Author):

my previous comments have been addressed, thank you very much and congratulations for the final manuscript.

Reviewer #3 (Remarks to the Author):

These are sufficient and comprehensive responses to my original reviewer comments, I have no further comments to add at this point.